# HQA-Attack: Toward High Quality Black-Box Hard-Label Adversarial Attack on Text

**Han Liu**
Dalian University of Technology
Dalian, China
liu.han.dut@gmail.com

**Zhi Xu**
Dalian University of Technology
Dalian, China
xu.zhi.dut@gmail.com

**Xiaotong Zhang***
Dalian University of Technology
Dalian, China
zxt.dut@hotmail.com

**Feng Zhang**
Peking University
Beijing, China
zfeng.maria@gmail.com

**Fenglong Ma**
The Pennsylvania State University
Pennsylvania, USA
fenglong@psu.edu

**Hongyang Chen**
Zhejiang Lab
Hangzhou, China
dr.h.chen@ieee.org

**Hong Yu**
Dalian University of Technology
Dalian, China
hongyu@dlut.edu.cn

**Xianchao Zhang***
Dalian University of Technology
Dalian, China
xczhang@dlut.edu.cn

## Abstract

Black-box hard-label adversarial attack on text is a practical and challenging task, as the text data space is inherently discrete and non-differentiable, and only the predicted label is accessible. Research on this problem is still in the embryonic stage and only a few methods are available. Nevertheless, existing methods rely on the complex heuristic algorithm or unreliable gradient estimation strategy, which probably fall into the local optimum and inevitably consume numerous queries, thus are difficult to craft satisfactory adversarial examples with high semantic similarity and low perturbation rate in a limited query budget. To alleviate above issues, we propose a simple yet effective framework to generate high quality textual adversarial examples under the black-box hard-label attack scenarios, named HQA-Attack. Specifically, after initializing an adversarial example randomly, HQA-attack first constantly substitutes original words back as many as possible, thus shrinking the perturbation rate. Then it leverages the synonym set of the remaining changed words to further optimize the adversarial example with the direction which can improve the semantic similarity and satisfy the adversarial condition simultaneously. In addition, during the optimizing procedure, it searches a transition synonym word for each changed word, thus avoiding traversing the whole synonym set and reducing the query number to some extent. Extensive experimental results on five text classification datasets, three natural language inference datasets and two real-world APIs have shown that the proposed HQA-Attack method outperforms other strong baselines significantly.

---

*Corresponding author.

37th Conference on Neural Information Processing Systems (NeurIPS 2023).

# 1   Introduction

Deep neural networks (DNNs) have achieved a tremendous success and are extremely popular in various domains, such as computer vision [34, 35], natural language processing [31, 15], robotics [2, 26] and so on. In spite of the promising performance achieved by DNN models, there are some concerns around their robustness, as evidence shows that even a slight perturbation to the input data can fool these models into producing wrong predictions [10, 19, 11, 5], and these perturbed examples are named as adversarial examples. Investigating the generation rationale behind adversarial examples seems a promising way to improve the robustness of neural networks, which motivates the research about adversarial attack. Most existing adversarial attack methods focus on computer vision [32, 36, 39] and have been well explored. However, the adversarial attack on text data is still challenging, as not only the text data space is intrinsically discrete and non-differentiable, but also changing words slightly may affect the fluency in grammar and the consistency in semantics seriously.

Based on the accessibility level of victim models, existing textual adversarial attack methods can be categorized into ***white-box attacks*** [9, 25, 7] and ***black-box attacks*** [19, 41, 20]. For white-box attacks, the attackers are assumed to have full information about the victim model, including training data, model architecture and parameters. Therefore, it is easy to formulate this type of attack as an optimization problem and utilize the gradient information to generate adversarial examples. However, as most model developers are impossible to release all the model and data information, white-box attacks seem excessively idealistic and cannot work well in real-world applications. For black-box attacks, the attackers are assumed to have only access to the predicted results, e.g., confidence scores or predicted labels, which seem more realistic. Existing black-box textual attack methods can be divided into ***soft-label setting*** [20, 6, 14] and ***hard-label setting*** [23, 38, 37]. For soft-label methods, they require the victim model to provide the confidence scores to calculate the importance of each word, and then replace words sequentially until an adversarial example is generated. However, it is also impractical as most real APIs do not allow users to access the confidence scores. For hard-label methods, they only need to know the predicted labels of the victim model to fulfill the attack task, thus are more practicable and promising.

Only a handful of methods are proposed to deal with the black-box hard-label textual adversarial attack task, which mainly rely on heuristic-based [23] or gradient-based strategies [38, 37]. Specifically, HLGA [23] is the first hard-label adversarial attack method, which leverages population-based heuristic optimization algorithm to craft plausible and semantically similar adversarial examples. However, as it requires a large population of adversarial candidates and population-based optimization strategy is easy to fall into the local optimum, HLGA inevitably consumes numerous query numbers. TextHoaxer [38] formulates the budgeted hard-label adversarial attack task on text data as a gradient-based optimization problem of perturbation matrix in the continuous word embedding space. LeapAttack [37] utilizes the gradient-based optimization by designing a novel mecha-

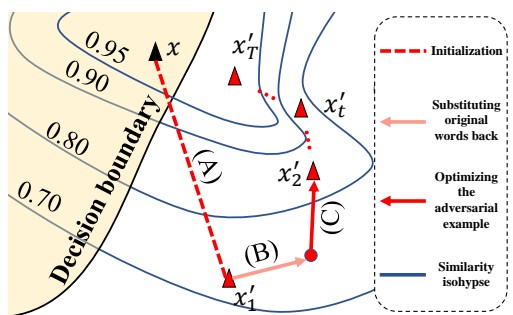

Figure 1: The overview of HQA-Attack. (A) Generate an adversarial example $x_1'$ by initialization. (B) Update $x_t'(1 \leqslant t \leqslant T)$ by substituting original words back. (C) Obtain $x_{t+1}'$ by optimizing the adversarial example. The similarity isohypse is a line which consists of points having the equal semantic similarity with the original example $x$.

nism that can interchange discrete substitutions and continuous vectors. Although these gradient-based methods improve the query efficiency to some extent, they still need some unnecessary queries caused by inaccurate gradient estimation. Furthermore, in tight budget scenarios, query inefficiency will directly bring about serious side effects on the semantic similarity and perturbation rate.

To alleviate the above issues, we propose a simple yet effective framework for producing **H**igh **Q**uality black-box hard-label **A**dversarial **Attack**, named **HQA-Attack**. The overview of HQA-Attack is shown in Figure 1. By "high quality", it means that the HQA-Attack method can generate adversarial examples with high semantic similarity and low perturbation rate under a tight query budget. Specifically, HQA-Attck first generates an adversarial example by initialization, and then

sequentially substitutes original words back as many as possible, thus shrinking the perturbation rate. Finally, it utilizes the synonym set to further optimize the adversarial example with the direction which can improve the semantic similarity and satisfy the adversarial condition simultaneously. In addition, to avoid going through the synonym set, it finds a transition synonym word for each changed word, thus reducing the query number to a certain extent. Experimental results on eight public benchmark datasets and two real-word APIs (Google Cloud and Alibaba Cloud) have demonstrated that HQA-Attack performs better than other strong baselines in the semantic similarity and perturbation rate under the same query budget. The source code and demo are publicly available at https://github.com/HQA-Attack/HQAAttack-demo.

## 2 Related Work

### 2.1 Soft-Label Textual Adversarial Attack

Adversarial attacks in soft-label settings rely on the probability distribution of all categories to generate adversarial examples. A series of strategies [16, 21, 24] utilize the greedy algorithm to craft adversarial examples, which first determine the word replacement order and greedily replace each word under this order. TextFooler [16] first determines the word replacement order according to the prediction change after deleting each word, and then replaces words back according to the word importance until adversarial examples are generated. Similarly, TextBugger [21] calculates the importance of sentences and the importance of words respectively by comparing the prediction before and after removing them. In addition, there are some methods [40, 1] which use combinatorial optimization algorithm to generate adversarial examples.

### 2.2 Hard-Label Textual Adversarial Attack

Adversarial attacks in hard-label settings only allow to access the predicted label, which seem more challenging and practical. HLGA [23] is the first hard-label textual adversarial attack method. It uses random initialization and search space reduction to get an incipient adversarial example, and then uses the genetic algorithm including mutation, selection and crossover three operations to further optimize the adversarial example. Although HLGA can generate adversarial examples with high semantic similarity with the original example and low perturbation rate, it needs to maintain a large candidate set in each iteration, which wastes a large number of queries. To alleviate this issue, TextHoaxer [38] uses the word embedding space to represent the text, and introduces a perturbation matrix and a novel objective function which consists of a semantic similarity term, a pair-wise perturbation constraint and a sparsity constraint. By using the gradient-based strategy to optimize the perturbation matrix, TextHoaxer can generate appropriate adversarial examples in a tight budget. LeapAttack [37] is another gradient-based method. After random initialization, it optimizes the adversarial example by constantly moving the example closer to the decision boundary, estimating the gradient and finding the proper words to update the adversarial example. In Appendix K, we further discuss some potential application scenarios about hard-label textual adversarial attack.

## 3 Problem Formulation

In this paper, we focus on the task of black-box hard-label textual adversarial attack, i.e., attackers can only access to the predicted label from the victim model to generate adversarial examples. Specifically, given an original example $x = [w_1, w_2, ..., w_n]$ with the ground truth label $y$, where $w_i$ is the $i$-th word, and $n$ is the total number of words in $x$. This task aims to construct an adversarial example $x' = [w'_1, w'_2, ..., w'_n]$ through replacing the original

Table 1: Symbol explanation.

| Symbol | Explanation |
|---|---|
| $x$ | the original example $x = [w_1, w_2, ..., w_n]$ |
| $x'$ | the adversarial example $x' = [w'_1, w'_2, ..., w'_n]$ |
| $x'_t$ | the adversarial example in the $t$-th step |
| $\bar{w}_i$ | the transition word associated with $w_i$ |
| $\boldsymbol{u}$ | the updating direction |
| $\boldsymbol{v}_{w_i}$ | the word vector of $w_i$ |
| $f$ | the victim model |
| $y$ | the true label of $x$ |
| $S(w_i)$ | the synonym set of $w_i$ |
| $x'(w_i)$ | the example after replacing the $i$-th word of $x'$ with $w_i$ |
| $Sim(\cdot, \cdot)$ | the similarity function between sentences |

word $w_i$ with a synonym $w'_i$ in the synonym set $S(w_i)$ which includes $w_i$ itself, thus misleading the victim model $f$ to output an incorrect prediction result:

$$f(x') \neq f(x) = y, \tag{1}$$

where Eq. (1) can be seen as the adversarial condition. There may exist several adversarial examples which can satisfy Eq. (1), but an optimal adversarial example $x^*$ is the one that has the highest semantic similarity with the original example $x$ among all the candidates. Formally,

$$x^* = \arg\max_{x'} Sim(x, x'), \text{ s.t. } f(x') \neq f(x), \tag{2}$$

where $Sim(x, x')$ is the semantic similarity between $x$ and $x'$. Table 1 summarizes the symbol explanation in detail.

## 4 The Proposed Strategy

### 4.1 Initialization

To fulfill the black-box hard-label textual adversarial attack task, we follow previous works [23, 38, 37] to first utilize random initialization to generate an adversarial example. Specifically, considering each word $w_i$ whose part-of-speech (POS) is *noun*, *verb*, *adverb* and *adjective* in the original example $x$, we randomly select a synonym $w'_i$ from the synonym set $S(w_i)$ of $w_i$ as the substitution of $w_i$, and repeat this procedure until the generated adversarial example $x'$ satisfies the adversarial condition.

Obviously, using random initialization to generate the adversarial example usually needs to change multiple words in the original example, thus necessarily leading to the low semantic similarity and large perturbation rate. To alleviate the above issue, we attempt to iteratively optimize the semantic similarity between the original example $x$ and the adversarial example in the $t$-th iteration $x'_t$, where $1 \leqslant t \leqslant T$ and $T$ is the total number of iterations. Furthermore, given an adversarial example $x'_t$, we can generate the adversarial example $x'_{t+1}$ by the following steps. (1) Substituting original words back; (2) Optimizing the adversarial example.

### 4.2 Substituting Original Words Back

In order to improve the semantic similarity of the generated adversarial example, previous works [23, 38, 37] first use the semantic similarity improvement brought by each original word as a measure to decide the replacement order, and then continually put original words back in each iteration. It means that in each iteration, by going through each word in the adversarial example, the semantic similarity improvement is only calculated once, and the replacement order is determined beforehand. However, making each replacement with the original word will change the intermediate generated adversarial example, thereby affecting the semantic similarity between the original example and the intermediate generated adversarial example, so just calculating the semantic similarity improvement at the beginning of each iteration to decide the replacement order is inaccurate.

To address the above problem, we propose to constantly substitute the original word which can make the intermediate generated adversarial example have the highest semantic similarity with the original example until the adversarial condition is violated. Specifically, given the original sample $x = [w_1, w_2, ..., w_n]$ and the adversarial example $x'_t = [w'_1, w'_2, ..., w'_n]$ in the $t$-th iteration, we can substitute original words back with the following steps.

1. Picking out an appropriate substitution word $w_*$ from $x$ with the following formula:

$$w_* = \arg\max_{w_i \in x} Sim(x, x'_t(w_i)) \cdot \mathcal{C}(f, x, x'_t(w_i)), \tag{3}$$

   where $Sim(x, x'_t(w_i))$ is the semantic similarity between $x$ and $x'_t(w_i)$, and $x'_t(w_i)$ is the example obtained by substituting the corresponding word $w'_i$ with $w_i$. $\mathcal{C}(f, x, x'_t(w_i))$ is a two-valued function defined as:

$$\mathcal{C}(f, x, x'_t(w_i)) = \begin{cases} 1, & f(x) \neq f(x'_t(w_i)) \\ 0, & f(x) = f(x'_t(w_i)) \end{cases}, \tag{4}$$

   where $f$ denotes the victim model. $\mathcal{C}$ equals to 1 if $f(x) \neq f(x'_t(w_i))$, and 0 otherwise.

2. If $\mathcal{C}(f, x, x_t'(w_*)) = 1$, it indicates that $x_t'(w_*)$ can attack successfully. We substitute the corresponding original word in $x_t'$ with $w_*$, and repeat the above step.

3. If $\mathcal{C}(f, x, x_t'(w_*)) = 0$, it indicates that $x_t'(w_*)$ cannot satisfy the adversarial condition. We terminate the swapping procedure, and return the result of the previous step.

After the above procedure, we can obtain a new adversarial example $x_t'$ which can retain the original words as many as possible, thus improving the semantic similarity and reducing the perturbation rate. The algorithm procedure is shown in Appendix A, and the analysis of computational complexity and query numbers is shown in Appendix C.1.

## 4.3 Optimizing the Adversarial Example

To further ameliorate the quality of the generated adversarial example, we optimize the adversarial example by leveraging the synonym set of each word. One may argue that we could directly traverse the synonym set to seek out the most ideal synonym word which has the highest semantic similarity and satisfies the adversarial condition simultaneously. However, in most real-world application scenarios, the query number is usually limited. To avoid going through the synonym set, we propose to optimize the adversarial example with the following two steps. (1) Determining the optimizing order; (2) Updating the adversarial example sequentially. The analysis of computational complexity and query numbers is shown in Appendix C.2.

### 4.3.1 Determining the Optimizing Order

In this step, we aim to determine a suitable optimizing order. To ensure the diversity of the generated adversarial example, we utilize the sampling method to determine the optimizing order. The probability distribution used by the sampling method is generated as follows. For $w_i'$ in $x_t'$ and $w_i$ in $x$, we first use the counter-fitting word vectors [27] to obtain their corresponding word vectors, and calculate the cosine distance between them as follows:

$$d_i = 1 - cos(\boldsymbol{v}_{w_i}, \boldsymbol{v}_{w_i'}), \tag{5}$$

where $\boldsymbol{v}_{w_i}$ and $\boldsymbol{v}_{w_i'}$ denote the word vectors of $w_i$ and $w_i'$ respectively. $cos(\cdot, \cdot)$ is the cosine similarity function. Then we compute the probability $p_i$ associated with the position of $w_i$ in $x$ with the following formula:

$$p_i = \frac{2 - d_i}{\sum_{j=1}^m (2 - d_j)}, \tag{6}$$

where $m$ is the total number of changed words between $x_t'$ and $x$. According to the probability distribution, we can obtain the optimizing order for $x_t'$.

### 4.3.2 Updating the Adversarial Example Sequentially

According to the optimizing order, we update the adversarial example with the synonym set sequentially. In particular, for the adversarial example $x_t'$ in the $t$-th iteration, we update it with the following steps. (1) Finding the transition word; (2) Estimating the updating direction; (3) Updating the adversarial example.

**Finding the transition word.** This step aims to search a reasonable transition word, thus avoiding traversing the synonym set for each changed word. Given the adversarial example $x_t' = [w_1', w_2', ..., w_n']$ and the current optimized word $w_i'$, we randomly select $r$ synonyms from $S(w_i)$ to construct the set $R = \{w_i^{(1)}, w_i^{(2)}, .., w_i^{(r)}\}$, use each element in $R$ to replace $w_i'$ in $x_t'$, and then obtain the transition word $\bar{w}_i$ with the following formula:

$$\bar{w}_i = \underset{w_i^{(j)} \in R}{\arg\max} \, Sim(x, x_t'(w_i^{(j)})) \cdot \mathcal{C}(f, x, x_t'(w_i^{(j)})), \tag{7}$$

where $x_t'(w_i^{(j)})$ is the example obtained by substituting the corresponding word $w_i'$ in $x_t'$ with $w_i^{(j)} \in R$. According to Eq. (7), we can get that $\bar{w}_i$ can make the example adversarial and improve the semantic similarity to some extent, while avoiding going through the synonym set. Furthermore, we can search other possible replacement words around the transition word $\bar{w}_i$.

**Estimating the updating direction.** As the transition word $\bar{w}_i$ originates from a randomly generated synonym set, we can further optimize it with a reasonable direction. Specifically, we first generate the set $\mathcal{K} = \{\bar{w}_i^{(1)}, \bar{w}_i^{(2)}..., \bar{w}_i^{(k)}\}$ by randomly sampling $k$ synonyms from $S(\bar{w}_i)$, and then obtain the set $\mathcal{M} = \{x_t'(\bar{w}_i^{(1)}), x_t'(\bar{w}_i^{(2)}), ..., x_t'(\bar{w}_i^{(k)})\}$, where $x_t'(\bar{w}_i^{(j)})$ is the example by replacing $w_i'$ in $x_t'$ with $\bar{w}_i^{(j)}$. By calculating the semantic similarity between each element in $\mathcal{M}$ and the original text $x$, we can get the set $\mathcal{S} = \{s^{(1)}, s^{(2)}, ..., s^{(k)}\}$, where $s^{(j)} = Sim(x, x_t'(\bar{w}_i^{(j)}))$ is the semantic similarity between $x$ and $x_t'(\bar{w}_i^{(j)})$. In the similar manner, the semantic similarity between $x$ and $x_t'(\bar{w}_i)$ can be computed $\bar{s}_i = Sim(x, x_t'(\bar{w}_i))$.

Intuitively, if $s^{(j)} - \bar{s}_i > 0$, it indicates that pushing the word vector $\boldsymbol{v}_{\bar{w}_i}$ towards $\boldsymbol{v}_{\bar{w}_i^{(j)}}$ tends to increase the semantic similarity, i.e., $\boldsymbol{v}_{\bar{w}_i^{(j)}} - \boldsymbol{v}_{\bar{w}_i}$ is the direction which can improve the semantic similarity. And if $s^{(j)} - \bar{s}_i < 0$, moving the word vector along the inverse direction of $\boldsymbol{v}_{\bar{w}_i^{(j)}} - \boldsymbol{v}_{\bar{w}_i}$ can improve the semantic similarity. Based on the above intuition, we estimate the final updating direction $\boldsymbol{u}$ by weighted averaging over the $k$ possible directions. Formally,

$$\boldsymbol{u} = \sum_{j=1}^{k} \alpha_j (\boldsymbol{v}_{\bar{w}_i^{(j)}} - \boldsymbol{v}_{\bar{w}_i}), \tag{8}$$

where $\alpha_j$ is the corresponding weight associated with the direction $\boldsymbol{v}_{\bar{w}_i^{(j)}} - \boldsymbol{v}_{\bar{w}_i}$, and it can be calculated by $\alpha_j = (s^{(j)} - \bar{s}_i) / \sum_{l=1}^{k} |s^{(l)} - \bar{s}_i|$.

**Updating the adversarial example.** Due to the discrete nature of text data, we need to use the updating direction $\boldsymbol{u}$ to pick out the corresponding replacement word $\widetilde{w}_i$ from $S(w_i)$, where $\widetilde{w}_i$ is the word which has the maximum cosine similarity between $\boldsymbol{u}$ and $\boldsymbol{v}_{\widetilde{w}_i} - \boldsymbol{v}_{\bar{w}_i}$ and ensures that $x_t'(\widetilde{w}_i)$ satisfies the adversarial condition. After obtaining $\widetilde{w}_i$, we can generate $x_{t+1}'$ in the optimizing order sequentially. In addition, to reduce the number of queries and shrink the perturbation rate, when implementing the program, we first use $x$ to initialize $x_{t+1}'$, and then replace the word $\widetilde{w}_i$ one by one until $x_{t+1}'$ satisfies the adversarial condition.

### 4.4 The Overall Procedure

The detailed algorithm procedure of HQA-Attack is given in Appendix B. In particular, HQA-Attack first gets the initial adversarial example by random initialization. Then it enters into the main loop. In each iteration, HQA-Attack first substitutes original words back, then determines the optimizing order, and finally updates the adversarial example sequentially. In addition, we provide some mechanism analysis of HQA-Attack from the perspective of decision boundary in Appendix D.

## 5 Experiments

### 5.1 Experimental Settings

**Datasets.** We conduct experiments on five public text classification datasets **MR** [28], **AG's News** [42], **Yahoo** [42], **Yelp** [42], **IMDB** [22], and three natural language inference datasets **SNLI** [3], **MNLI** [33], **mMNLI** [33]. The detailed dataset description is shown in Appendix E. We follow the previous methods [23, 38, 37] to take 1000 test examples of each dataset to conduct experiments.

**Baselines.** We compare with three state-of-the-art black-box hard-label textual adversarial attack methods: (1) **HLGA** [23] is a hard-label adversarial attack method that employs the genetic algorithm to generate the adversarial example. (2) **TextHoaxer** [38] is a hard-label adversarial attack method that formulates the budgeted hard-label adversarial attack task on text data as a gradient-based optimization problem of perturbation matrix in the continuous word embedding space. (3) **LeapAttack** [37] is a recent hard-label adversarial attack method, which estimates the gradient by the Monte Carlo method.

**Evaluation Metrics.** We use two widely used evaluation metrics *semantic similarity* and *perturbation rate*. For semantic similarity, we utilize the universal sequence encoder [4] to calculate the semantic similarity between two texts. The range of the semantic similarity is between $[0, 1]$, and the larger semantic similarity indicates the better attack performance. For perturbation rate, we use the ratio

of the number of changed words over the number of total words in the adversarial example, and the lower perturbation rate indicates the better results.

**Victim Models.** We follow [23, 38, 37] to adopt three widely used natural language processing models as victim models: **BERT** [8], **WordCNN** [17], and **WordLSTM** [13]. All the model parameters are taken from the previous works [23, 38, 37]. We also attack some advanced models like T5 [30] and DeBERT [12], and the results are shown in Appendix F. To further verify the effectiveness of different algorithms in real applications, we also attempt to use **Google Cloud API** (https://cloud.google.com/natural-language) and **Alibaba Cloud API** (https://ai.aliyun.com/nlp) as the victim models.

**Implementation Details.** For the random initialization, we employ the same method used in previous methods [23, 38, 37]. After the initialization, we follow [23, 38] to perform a pre-processing step to remove the unnecessary replacement words. For the hyperparameters, we consistently set $r = 5$ and $k = 5$ for all the datasets. The detailed parameter investigation is provided in Appendix G. In addition, during the optimization procedure, if we re-optimize the same adversarial example three times and no new better adversarial examples are generated, we randomly go back to the last three or four adversarial example. We do not re-optimize an adversarial example more than two times. For fair comparison, we follow [23, 38, 37] to generate 50 synonyms for each word by using the counter-fitting word vector. We also conduct experiments based on BERT-based synonyms, and the results are shown in Appendix H.

## 5.2 Experimental Results

### 5.2.1 Comparison on Semantic Similarity and Perturbation Rate

We exactly follow the previous work [38] to set the query budget to 1000, i.e., the number of allowed queries from the attacker is 1000. As different algorithms use the same random initialization step which determines the prediction accuracy after the adversarial attack, so different algorithms have the same prediction accuracy. Our goal is to generate the adversarial examples with higher semantic similarity and lower perturbation rate. Tables 2 and 3 report the experimental results when attacking text classification models. The best results are highlighted in bold.

As shown in Tables 2 and 3, when the query limit is 1000, for different datasets and tasks, HQA-Attack can always generate adversarial examples that have the highest semantic similarity and the lowest perturbation rate. Specifically, for the dataset MR with short text data, HQA-Attack increases the average semantic similarity by 6.9%, 6.5%, 6.9% and decreases the average perturbation rate by 0.777%, 0.832%, 0.983% compared with the second best method when attacking BERT, WordCNN and WordLSTM respectively. For the dataset IMDB with long text data, HQA-Attack increases the average semantic similarity by 4.5%, 3.4%, 2.7% and decreases the average perturbation rate by 1.426%, 0.601%, 0.823% compared with the second best method when attacking BERT, WordCNN and WordLSTM respectively. For the dataset with more than two categories like AG, HQA-Attack increases the average semantic similarity by 10.6%, 8.8%, 11.6% and decreases the average perturbation rate by 4.785%, 3.885%, 5.237% compared with the second best method when attacking BERT, WordCNN and WordLSTM respectively. All these results demonstrate that HQA-Attack can generate high-quality adversarial examples in the tight-budget hard-label setting.

### 5.2.2 Comparison on Attack Efficiency

The attack efficiency is an important criterion in evaluating the attack performance, as in most DNN-based NLP platforms the number of queries is limited. Therefore, We further compare the proposed HQA-Attack with two latest methods TextHoaxer and LeapAttack under different query budgets $[100, 300, 500, 700, 1000]$ on text classification datasets. As shown in Figure 2, with the query budget increasing, the average semantic similarity of all the methods keeps increasing and the average perturbation rate of all the methods keeps decreasing. In terms of semantic similarity and perturbation rate, HQA-Attack always performs much better than other methods in all the budgets. These results further validate that our proposed HQA-Attack has the ability to generate adversarial examples with higher semantic similarity and lower perturbation rate in different budget limits.

### 5.2.3 Attack Real-World APIs

Table 2: Comparison of semantic similarity (Sim) and perturbation rate (Pert) with the budget limit of 1000 when attacking text classification models. Acc stands for model prediction accuracy after the adversarial attack, which is determined by the random initialization step and the same for different adversarial attack models.

| Dataset | Method | BERT | | | WordCNN | | | WordLSTM | | |
|---|---|---|---|---|---|---|---|---|---|---|
| | | Acc(%) | Sim(%) | Pert(%) | Acc(%) | Sim(%) | Pert(%) | Acc(%) | Sim(%) | Pert(%) |
| MR | HLGA | 1.0 | 62.5 | 14.532 | 0.7 | 64.4 | 14.028 | 0.7 | 63.5 | 14.462 |
| | TextHoaxer | | 67.3 | 11.905 | | 68.6 | 12.056 | | 67.3 | 12.324 |
| | LeapAttack | | 61.6 | 14.643 | | 63.2 | 14.016 | | 61.3 | 14.435 |
| | HQA-Attack | | **74.2** | **11.128** | | **75.1** | **11.224** | | **74.2** | **11.341** |
| AG | HLGA | 2.8 | 60.5 | 17.769 | 1.4 | 71.9 | 13.855 | 5.7 | 61.8 | 17.890 |
| | TextHoaxer | | 63.2 | 15.766 | | 73.9 | 12.716 | | 63.8 | 16.520 |
| | LeapAttack | | 62.6 | 16.143 | | 72.0 | 12.827 | | 63.0 | 17.028 |
| | HQA-Attack | | **73.8** | **10.981** | | **82.7** | **8.831** | | **75.4** | **11.283** |
| Yahoo | HLGA | 0.8 | 68.7 | 7.453 | 0.8 | 71.9 | 8.564 | 1.9 | 63.8 | 9.531 |
| | TextHoaxer | | 70.2 | 6.841 | | 74.8 | 7.740 | | 67.0 | 8.502 |
| | LeapAttack | | 66.7 | 7.448 | | 74.3 | 7.842 | | 64.7 | 9.095 |
| | HQA-Attack | | **76.4** | **5.609** | | **82.4** | **6.132** | | **73.9** | **6.645** |
| Yelp | HLGA | 0.6 | 71.9 | 10.411 | 0.6 | 79.7 | 9.102 | 3.2 | 78.8 | 8.654 |
| | TextHoaxer | | 73.8 | 9.585 | | 81.3 | 8.545 | | 80.4 | 8.108 |
| | LeapAttack | | 72.7 | 9.877 | | 80.1 | 8.816 | | 79.6 | 8.111 |
| | HQA-Attack | | **81.9** | **6.756** | | **87.8** | **6.312** | | **86.7** | **5.786** |
| IMDB | HLGA | 0.1 | 83.2 | 5.571 | 0.0 | 87.6 | 4.464 | 0.3 | 87.6 | 4.464 |
| | TextHoaxer | | 84.7 | 5.202 | | 88.8 | 4.197 | | 88.8 | 4.197 |
| | LeapAttack | | 84.0 | 5.041 | | 89.7 | 3.886 | | 89.0 | 4.021 |
| | HQA-Attack | | **89.2** | **3.615** | | **93.1** | **3.285** | | **91.7** | **3.198** |

Table 3: Comparison of semantic similarity (Sim) and perturbation rate (Pert) with the budget limit of 1000 when attacking the natural language inference model (BERT).

| Method | SNLI | | | MNLI | | | mMNLI | | |
|---|---|---|---|---|---|---|---|---|---|
| | Acc(%) | Sim(%) | Pert(%) | Acc(%) | Sim(%) | Pert(%) | Acc(%) | Sim(%) | Pert(%) |
| HLGA | 1.3 | 35.9 | 18.510 | 2.9 | 49.6 | 14.498 | 1.7 | 50.7 | 14.349 |
| TextHoaxer | | 38.7 | 16.615 | | 52.9 | 12.730 | | 54.4 | 12.453 |
| LeapAttack | | 35.0 | 19.905 | | 49.1 | 15.728 | | 50.2 | 15.135 |
| HQA-Attack | | **54.2** | **15.958** | | **64.7** | **12.093** | | **65.4** | **11.502** |

To further verify the effectiveness of different algorithms, we attempt to use TextHoaxer, LeapAttack and HQA-Attack to attack two real-world APIs: Google Cloud (https://cloud.google.com/natural-language) and Alibaba Cloud (https://ai.aliyun.com/nlp). To further evaluate the fluency of the generated

Table 4: Comparison of semantic similarity and perturbation rate when attacking against real-world APIs.

| API | Google Cloud | | | Alibaba Cloud | | |
|---|---|---|---|---|---|---|
| | Sim(%) | Pert(%) | PPL | Sim(%) | Pert(%) | PPL |
| TextHoaxer | 76.1 | 7.179 | 253 | 78.3 | 6.190 | 261 |
| LeapAttack | 73.2 | 10.699 | 295 | 77.7 | 7.198 | 285 |
| HQA-Attack | **81.8** | **7.117** | **244** | **83.4** | **6.183** | **255** |

adversarial examples, we add the perplexity (PPL) as the additional evaluation metric which is calculated by using GPT-2 Large [29]. The lower PPL indicates the better performance. As Google and Alibaba only provide limited service budgets, we select 100 examples from the MR dataset whose lengths are greater than or equal to 20 words to perform experiments, and restrict that each method can only query the API 350 times. Table 4 shows the results of TextHoaxer, LeapAttack and HQA-Attack. It can be seen that compared with the second best results, HQA-Attack increases the semantic similarity 5.7%, 5.1%, decreases the perturbation rate 0.062%, 0.007% and decreases the PPL 9, 6 on Google Cloud and Alibaba Cloud respectively. We also compare the performance of TextHoaxer, LeapAttack and HQA-Attack in different budget limits. The results are shown in Figure

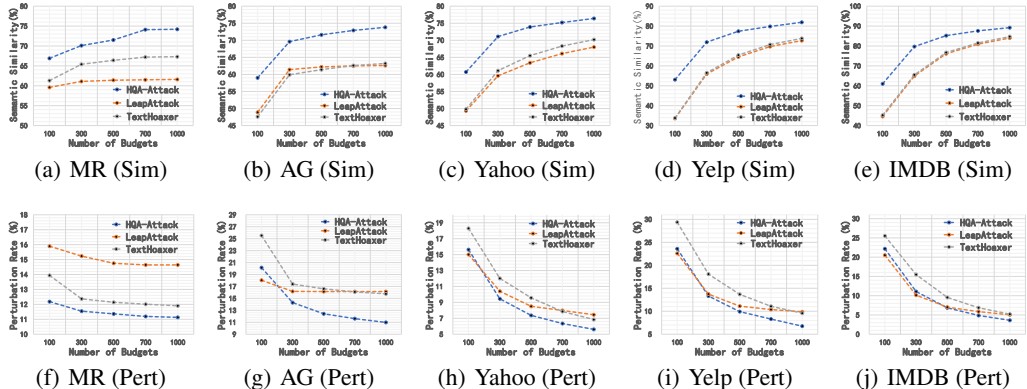

Figure 2: Comparison on semantic similarity and perturbation rate in different budget limits when attacking against BERT.

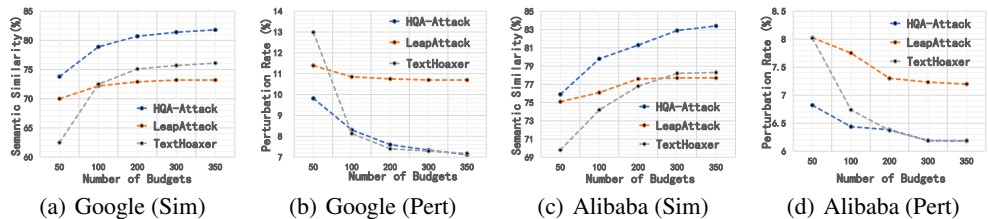

Figure 3: Comparison of semantic similarity and perturbation rate in different budget limits when attacking against real-world APIs.

Table 5: Human evaluation results in average classification accuracy(%).

| Dataset | HLGA | TextHoaxer | LeapAttack | HQA-Attack |
|---------|------|------------|------------|------------|
| MR | 82.6 | 84.8 | 84.0 | **87.4** |
| IMDB | 84.4 | 85.4 | 85.0 | **88.6** |

3. We can get that HQA-Attack can have the higher semantic similarity and lower perturbation rate in most cases, which further demonstrates the superiority of HQA-Attack over other baselines.

### 5.2.4 Human Evaluation

We have conducted the human evaluation experiments on the BERT model using HLGA, TextHoaxer, LeapAttack and HQA-Attack for the MR and IMDB datasets. Specifically, for each dataset, we first randomly select 50 original samples, and use each adversarial attack method to generate the corresponding 50 adversarial examples respectively. Then we ask 10 volunteers to annotate the class labels for these samples, and calculate the average classification accuracy (Acc) for each method. Intuitively, if the accuracy is higher, it means that the quality of the generated adversarial examples is better. The Acc(%) results of clean examples are 94.2% and 93.6% for MR and IMDB respectively. And the results of adversarial examples are shown in Table 5. The results show that the adversarial examples generated by HQA-Attack are more likely to be classified correctly, which further verifies the superiority of HQA-Attack in preserving the semantic information.

### 5.2.5 Attack Models which Defend with Adversarial Training

With the development of the AI security, a lot of works focus on defending against adversarial attacks. We further compare the attack performance when the victim model is trained with three effective adversarial training strategies **HotFlip** [9], **SHIELD** [18] and **DNE** [43]. We select the BERT model as the victim model, set the query budget to 1000 and then perform experiments on the AG dataset. We also use the perplexity (PPL) as the additional evaluation metric to judge the fluency

Table 6: Comparison results of attacking models which defend with adversarial training.

| Method | HotFlip | | | | SHIELD | | | | DNE | | | |
|---|---|---|---|---|---|---|---|---|---|---|---|---|
| | Acc(%) | Sim(%) | Pert(%) | PPL | Acc(%) | Sim(%) | Pert(%) | PPL | Acc(%) | Sim(%) | Pert(%) | PPL |
| HLGA | | 55.8 | 17.185 | 538 | | 59.8 | 15.206 | 565 | | 58.6 | 14.132 | 444 |
| TextHoaxer | 21.7 | 55.5 | 17.145 | 463 | 12.6 | 67.8 | 12.312 | 469 | 18.6 | 66.5 | 11.517 | 363 |
| LeapAttack | | 62.0 | 17.288 | 530 | | 66.3 | 14.868 | 541 | | 65.7 | 13.541 | 414 |
| HQA-Attack | | **72.7** | **10.192** | **350** | | **76.0** | **10.085** | **361** | | **75.0** | **9.808** | **312** |

Table 7: Ablation study.

| Dataset | Acc(%) | Random Initialization | | w/o Substituting | | w/o Optimizing | | HQA-Attack | |
|---|---|---|---|---|---|---|---|---|---|
| | | Sim(%) | Pert(%) | Sim(%) | Pert(%) | Sim(%) | Pert(%) | Sim(%) | Pert(%) |
| MR | 0.7 | 18.2 | 39.234 | 74.4 | 12.638 | 74.2 | 11.673 | **75.1** | **11.224** |
| AG | 1.4 | 30.5 | 43.463 | 80.2 | 16.519 | 79.3 | 11.305 | **82.7** | **8.831** |
| Yahoo | 0.8 | 28.0 | 32.143 | 73.1 | 15.982 | 79.7 | 7.445 | **82.4** | **6.132** |
| Yelp | 0.6 | 18.3 | 38.595 | 71.6 | 19.342 | 85.2 | 7.883 | **87.8** | **6.312** |
| IMDB | 0.0 | 35.0 | 30.963 | 72.9 | 17.229 | 92.2 | 3.923 | **93.1** | **3.285** |

of the generated adversarial examples. Table 6 shows the attack performance. We can observe that HQA-Attack can also obtain the best results compared with other strong baselines.

#### 5.2.6 Ablation Study and Case Study

To investigate the effectiveness of different components, we make the ablation study on five text classification datasets when attacking WordCNN. The results are shown in Table 7. Random Initialization means the adversarial examples generated only by the random initialization step. w/o Substituting means that the HQA-Attack model without the substituting original words back step. w/o Optimizing means that the HQA-Attack model which randomly selects a word that can keep the example adversarial as the replacement after substituting original words back without optimizing the adversarial example. It is easy to find that all modules contribute to the model, which verifies that the substituting original words back step is useful and the optimizing the adversarial example step is also indispensable. To further demonstrate the effectiveness of our proposed word back-substitution strategy, we add some extra experiments in Appendix I. We also list some concrete adversarial examples generated by HQA-Attack, which are shown in Appendix J. These examples further demonstrate that our proposed HQA-Attack model can generate a high-quality black-box hard-label adversarial example with only a small perturbation.

## 6 Conclusion

In this paper, we propose a novel approach named HQA-Attack for crafting high quality textual adversarial examples in black-box hard-label settings. By substituting original words back, HQA-Attack can reduce the perturbation rate greatly. By utilizing the synonym set of the remaining changed words to optimize the adversarial example, HQA-Attack can improve the semantic similarity and reduce the query budget. Extensive experimental results demonstrate that the proposed HQA-Attack method can generate high quality adversarial examples with high semantic similarity, low perturbation rate and fewer query numbers. In future work, we plan to attempt more optimization strategies to refine the model, thus further boosting the textual adversarial attack performance.

## Acknowledgments and Disclosure of Funding

The authors are grateful to the anonymous reviewers for their valuable comments. This work was supported by National Natural Science Foundation of China (No. 62106035, 62206038, 61972065) and Fundamental Research Funds for the Central Universities (No. DUT20RC(3)040, DUT20RC(3)066), and supported in part by Key Research Project of Zhejiang Lab (No. 2022PI0AC01) and National Key Research and Development Program of China (2022YFB4500300).

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
