# OpenReview forum: "HQA-Attack: Toward High Quality Black-Box Hard-Label Adversarial Attack on Text"
_NeurIPS.cc/2023/Conference — NeurIPS 2023 poster_

### Official Review · Reviewer_BWcn · 2023-07-02

**Soundness:** 2 fair
**Presentation:** 3 good
**Contribution:** 3 good
**Rating:** 6
**Confidence:** 3

**Summary:**

This paper proposed a new approach HQA-Attack using iterative substituting normal words and replacing with synonyms for creating high quality adversarial text samples in the setting of black box attack + hard label. The experimental results on several NLP datasets and two commercial APIs validate the effectiveness of the proposed methods.

**Strengths:**

1. This paper is well-written. The problem this paper focuses on is important, and the proposed method is interesting.
2. The experiments are sufficient, and the conclusion is convincing.


**Weaknesses:**

1. In this paper, the theoretical analysis (e.g., theorems) is missing. I think it is a suggested part, especially for a top machine learning conference like NeurIPS. More specifically, your proposed method still needs many iterations including calling functions to get hard labels. Even though empirical results shown in Figure 2, the reason about why the proposed method reduces the number of queries. A theriacal analysis about it is suggested. The analysis of computational complexity is also suggested especially the proposed method is described as practical.
2. The fluency of sentences seems not to be considered. When we replace the words in the sentence, we should consider if the remaining sentence is still fluent. Correct me if I am wrong.


**Questions:**

1. Just curious. Did authors consider using ChatGPT to help you generate some similar sentences?

**Limitations:**

The authors provided the description of limitations such as potential negative societal impact of their work.

---

> ### Author Rebuttal · Authors · 2023-08-09
>
> **For Reviewer BWcn.**
>
> Thanks very much for your valuable and positive comments.
>
> **1. Theoretical analysis of computational complexity and the number of queries.**
>
> We have attempted to analyze the computational complexity and the number of queries for the main procedures of HQA-Attack, including Section 4.2 "Substituting original words back" and Section 4.3 "Optimizing the adversarial examples".
>
> Given the original example $x$ and its initial adversarial example $x'$, $n$ represents the length of the original sample $x$. We can make the following analysis.
>
> (1) Analysis of substituting original words back.
>
> In the best case, $x'$ only has one different word compared with $x$, so Alg. 1 (Appendix A) only calculates the cosine similarity once and queries the victim model once. In this case, the computational complexity is $\mathcal{O}(1)$ and the number of queries is $\mathcal{O}(1)$.
>
> In the worst case, $x'$ has $n$ different words with $x$, which means we will conduct $n$ replacement iterations (Alg. 1 Line 1-Line 21). In this case, the number of calculating cosine similarity is $n, n-1, ..., 1$ from the $1$-st to the $n$-th iteration, so the computational complexity is $\frac{(n+1)\times n}{2} = \mathcal{O}(n^2)$. And for the number of queries, there are two boundary cases. For Alg. 1 Line 10-Line 17, we may successfully replace the first word or replace the last word in $choices$. In the former case, the number of queries is $n = \mathcal{O}(n)$. In the latter case, the number of queries in each iteration is $n, n-1, ..., 1$ from the $1$-st to the $n$-th iteration, so the number of queries is $\frac{(n+1)\times n}{2} = \mathcal{O}(n^2)$.
>
> According to the above analysis, for Section 4.2, the overall computational complexity is $\mathcal{O}(n^2)$ and the number of queries is $\mathcal{O}(n^2)$, where $n$ is the length of the original sample $x$.
>
>
> (2) Analysis of optimizing the adversarial examples.
>
> Assume that $r$ is the number of synonyms in finding the transition word (Line 199), $k$ is the number of synonyms in estimating the updating direction (Line 208), $C = |S(w_i)|$ is the synonym set size for $w_i$. For the sake of analysis, we assume the synonym set size of each word is the same.
>
> In Section 4.3.1 (Line 183), the number of calculating cosine similarity is $\mathcal{O}(n)$ and the number of queries is $0$. Specifically, $x'_t$ may have only one different word with the original sentence $x$, and also may have $n$ different words, thus the corresponding computational complexity is $\mathcal{O}(n)$, where $x'_t$ is the generated adversarial example through Section 4.2. And this step (Section 4.3.1) does not need to query the victim model.
>
> In Section 4.3.2 (Line 194), the number of calculating cosine similarity is $\mathcal{O}(n(r+k+C))$ and the number of queries is $\mathcal{O}(n(r+C))$. Specifically, In Alg. 2 (Appendix B) Line 9-Line 14, the number of $I$ may be $1$ or $n$. In each iteration (Alg. 2 Line 10-Line 13), for the first sub-step (Alg. 2 Line 10), we randomly select $r$ synonyms to find the transition word, so the computational complexity is $\mathcal{O}(r)$ and the number of queries is $\mathcal{O}(r)$. For the second sub-step (Alg. 2 Line 11), we randomly select $k$ synonyms to obtain the updating direction, so the computational complexity is $\mathcal{O}(k)$ and the number of queries is $0$. In the third sub-step (Alg. 2 Line 12), we obtain $\widetilde{w}_i$ from $S(w_i)$, so the computational complexity is $\mathcal{O}(C)$ and the number of queries is $\mathcal{O}(C)$. Furthermore, in the experiments, we set the size of the synonym set is 50, so $r$, $k$ and $C$ are all integers which are not greater than 50. Then both $\mathcal{O}(n(r+k+C))$ and $\mathcal{O}(n(r+C))$ can be simplified to $\mathcal{O}(n)$.
>
> According to the above analysis, for Section 4.3, the overall computational complexity is $\mathcal{O}(n)$ and the number of queries is $\mathcal{O}(n)$, where $n$ is the length of the original sample $x$.
>
> We will follow your comments to add the above analysis in the final version.
>
> **2. About the fluency of sentences.**
>
> We have evaluated the fluency of generated adversarial examples using the perplexity metric (PPL), which is a widely used metric for evaluating the sentence fluency [1,2,3]. In the original paper, we have provided the perplexity results in Tables 4 and 5, where the lower PPL values indicate the better fluency. It can be seen that our proposed HQA-Attack outperforms other baselines in terms of sentence fluency.
>
> [1] A Unified Evaluation of Textual Backdoor Learning: Frameworks and Benchmarks. NeurIPS 2022.
>
> [2] Subword Evenness (SuE) as a Predictor of Cross-lingual Transfer to Low-resource Languages. EMNLP 2022.
>
> [3] Gradient-based Constrained Sampling from Language Models. EMNLP 2022.
>
> **3. About using ChatGPT.**
>
> In our upcoming work, we will attempt to leverage ChatGPT or other large language models (LLMs) to help us generate and optimize the adversarial examples. Here are some possible points.
>
> (1) Using LLMs to generate similar words, phrases or sentences for the original text.
>
> (2) Taking the LLMs as the substitution model to imitate the victim model, thus reducing the number of queries.
>
> (3) Investigating how to design appropriate prompts to guide LLMs to generate the adversarial examples directly.

---

> > ### Comment · Area_Chair_sGUZ · 2023-08-19
> > **Thanks for your detailed response**
> >
> > Dear authors,
> >
> > Thanks for your detailed response. I think your response has addressed some of the reviewer's concerns.
> >
> > Best,
> >
> > AC

---

> > > ### Author Response · Authors · 2023-08-19
> > >
> > > Dear AC,
> > >
> > > We would like to thank you and each reviewer for devoting the valuable time and effort to assist us in enhancing the quality of our manuscript.
> > >
> > > Best regards,
> > >
> > > Authors

---

### Official Review · Reviewer_pVxP · 2023-07-03

**Soundness:** 2 fair
**Presentation:** 3 good
**Contribution:** 3 good
**Rating:** 6
**Confidence:** 5

**Summary:**

This paper presents a black-box adversarial attack method that can create high-quality adversarial samples without accessing the predictive probability/gradients of the victim model. The approach is based on synonyms substitution. In the attack framework, this paper also proposes some tricks to reduce the query number and increase the semantic similarity. Experimental results demonstrate the effectiveness of the proposed attack, indicating further attention is needed to look into this adversarial problem.

**Strengths:**

This paper looks into an important and interesting research problem. Black-box textual adversarial attack is a challenging problem due to the discrete nature of the text. This paper presents a concrete approach that can effectively address this problem, crafting high-quality adversarial samples with high semantic similarity and fewer query numbers.  Experimental results show that this approach is more effective than previous approaches, like LeapAttack.

**Weaknesses:**

Please correct me if I have some misunderstanding of the paper.

1. I don't see too much novelty in the proposed approach compared to previous work [1]. The basic intuitions are both first constructing a successful but low-quality adversarial sample, and then iteratively performing multiple rounds of substitution to improve the quality. If I understand correctly, only the reverse searching process is optimized in this paper.

2. I have some problems with the practical issues of the proposed approach and experimental settings, as indicated in this paper [2]. First, why do attackers want to attack a new classification model, or the sentiment analysis model you adopt in the experiments? What is the motivation and what benefits can they get? Please justify the reasonability of the considered experimental settings. Second, why do the attackers want to craft some high-quality samples that retain high similarity compared to the original ones? The goal of attacking is to bypass the detection system and retain only the adversarial meaning. So what is the motivation of the attackers to spend a huge amount of time doing engineering to retain the meaning of the whole sentence to the maximum?

3. For the evaluation, human annotation is important to verify the validity of adversarial samples, since many adversarial samples may be invalid due to the change in core semantics meaning, but they still retain high similarity computed by neural models.

4. In the experiments, only some old-school models are considered, such as BERT, WordCNN, LSTM. The results from these models don't have too much significance since none of them are frequently employed nowadays. The experiments should be conducted on some more advanced models, like T5, DeBERTa, and also may consider some other paradigms, like few-shot, and zero-shot inference via GPT.



[1] Generating Natural Language Attacks in a Hard Label Black Box Setting. Rishabh Maheshwary, Saket Maheshwary, Vikram Pudi. AAAI
[2] Why Should Adversarial Perturbations be Imperceptible? Rethink the Research Paradigm in Adversarial NLP. Yangyi Chen, Hongcheng Gao, Ganqu Cui, Fanchao Qi, Longtao Huang, Zhiyuan Liu, Maosong Sun. EMNLP

**Questions:**

Please see the second point in the Weakness section.

**Limitations:**

This paper has a limitation section that effectively addresses the limitations of the proposed approach.

---

> ### Author Rebuttal · Authors · 2023-08-09
>
> **For Reviewer pVxP.**
>
> Thanks very much for your valuable and helpful comments.
>
> **1. About the contributions.**
>
> We would like to clarify our contributions.
>
> (1) We propose a novel method for substituting original words back (Sec. 4.2), which is different from previous methods. Specifically, previous works only calculate the semantic similarity gain once and then determine the replacement order. However, the initially determined order is inaccurate after the adversarial example goes through several replacements. To solve this issue, we propose a new method which is shown in Alg. 1 (Appendix A). After replacing one original word successfully (Line 12-Line 13), our method can make the $flag$ = False (Line 14) and break out of the current loop (Line 15), where the current loop is from Line 10 to Line 17. Furthermore, as the $flag$ = False, our method will continue the outer loop and recompute the order, where the outer loop is from Line 1 to Line 21. By replacing the best original word in each iteration, HQA-Attack can make the intermediate generated adversarial example have the highest semantic similarity with the original example.
>
> (2) We propose a novel optimization strategy (Sec. 4.3), which is different from previous methods. Specifically, previous works rely on the complex heuristic algorithm or unreliable gradient estimation strategy, which probably fall into the local optimum and consume numerous queries. To alleviate above issues, in the optimizing procedure, HQA-Attack optimizes the adversarial example with the direction which can improve the semantic similarity and satisfy the adversarial condition simultaneously. And in the optimizing procedure, HQA-Attack searches a transition word for each changed word, thus reducing the queries to some extent.
>
> (3) Experimental results on 8 public datasets and 2 real-world APIs have shown that HQA-Attack outperforms other strong baselines greatly.
>
> **2. About the practical issues.**
>
> (1) The experimental settings are reasonable.
>
> First, the adversarial attack research aims to explore the system vulnerability rather than cause some destruction, and the classification model has been widely used in many real systems. As many APIs do not release the model and only provide the hard-label prediction results, so the hard-label black-box experimental settings are suitable for simulating such scenarios.
>
> Second, we have conducted experiments on the classification models provided by Alibaba Cloud API and Google Cloud API, which are both real applications. After identifying the system vulnerability, the potential application is to use the adversarial examples to improve these systems and mitigate some negative impacts.
>
> In addition, the experimental settings have been widely adopted by previous methods like HLGA, TextHoaxer and LeapAttack.
>
> (2) High-quality adversarial examples are essential.
>
> In real world applications, low-quality adversarial examples can be easily detected by manual inspection or detection algorithms, which are meaningless to improve the system robustness. High-quality adversarial examples have higher semantic similarity and lower perturbation rate, which are easier to bypass manual inspection and detection algorithms, thus fulfilling the attack task. Furthermore, by utilizing high-quality adversarial examples to improve the original system, we can make the system more robust and safer.
>
> **3. About human evaluation.**
>
> We have conducted the human evaluation experiments on the Bert model using HLGA, TextHoaxer, LeapAttack and HQA-Attack for the MR and IMDB datasets. Specifically, for each dataset, we first randomly select 50 original samples, and use each adversarial attack method to generate the corresponding 50 adversarial examples respectively. Then we ask 10 volunteers to annotate the class labels for these samples, and calculate the average classification accuracy (Acc) for each method. Intuitively, if the accuracy is higher, it means that the quality of the generated adversarial examples is better. The detailed Acc(%) results are as follows.
>
> | Dataset | HLGA | TextHoaxer | LeapAttack | HQA-Attack |
> | ------- | ---- | ---------- | ---------- | ---------- |
> | MR      | 82.6 | 84.8       | 84.0       | **87.4**   |
> | IMDB    | 84.4 | 85.4       | 85.0       | **88.6**   |
>
> The results show that the adversarial examples generated by HQA-Attack are more likely to be classified correctly, which further verifies the superiority of HQA-Attack in preserving the semantic information. We will add these results in the final version.
>
> **4. About experiments on advanced models.**
>
> We have conducted experiments to attack T5 and DeBERTa. Due to time limit, we select 1000 samples from each of sentence-level datasets (MR, AG and Yahoo) and 500 samples from document-level datasets (Yelp and IMDB). The results (Sim(%)/Pert(%)) are as follows.
>
> | Dataset (Method)           | T5          | DeBERTa     |
> | ----------------- | ----------- | ----------- |
> | MR (TextHoaxer)    | 65.7/12.463 | 66.6/12.251 |
> | MR (LeapAttack)    | 60.0/17.240 | 60.9/17.853 |
> | MR (HQA-Attack)    | **73.0/11.846** | **73.8/11.460** |
> | AG (TextHoaxer)    | 67.3/14.702 | 63.3/17.385 |
> | AG (LeapAttack)    | 65.7/17.697 | 62.7/20.289 |
> | AG (HQA-Attack)    | **76.8/11.365** | **75.0/13.019** |
> | Yahoo (TextHoaxer) | 64.7/7.735  | 66.0/9.360  |
> | Yahoo (LeapAttack) | 62.9/9.704 | 64.1/11.221 |
> | Yahoo (HQA-Attack) | **69.0/10.428**  | **72.2/10.051**  |
> | Yelp (TextHoaxer)  | 77.5/8.532  | 68.6/13.798 |
> | Yelp (LeapAttack)  | 76.3/10.348 | 68.8/15.370 |
> | Yelp (HQA-Attack)  | **84.1/7.514**  | **78.9/11.749** |
> | IMDB (TextHoaxer)  | 83.3/6.477  | 78.1/10.153 |
> | IMDB (LeapAttack)  | 83.7/7.087  | 78.7/11.018 |
> | IMDB (HQA-Attack)  | **87.0/6.142**  | **84.3/9.963**  |
>
>
> From the results, it can be seen that HQA-Attack still outperforms other strong baselines, which further verifies the superiority of HQA-Attack. We will add these results in the final version.

---

> > ### Comment · Reviewer_pVxP · 2023-08-17
> >
> > Thanks for your added experimental results. They seem to support the conclusion in the paper. However, I would like to discuss a little bit more about the practical issues of the proposed method.
> >
> >
> >
> > # Author Response
> > (1)  The experimental settings are reasonable.
> >
> > First, the adversarial attack research aims to explore the system vulnerability rather than cause some destruction, and the classification model has been widely used in many real systems. As many APIs do not release the model and only provide the hard-label prediction results, so the hard-label black-box experimental settings are suitable for simulating such scenarios.
> >
> > Second, we have conducted experiments on the classification models provided by Alibaba Cloud API and Google Cloud API, which are both real applications. After identifying the system vulnerability, the potential application is to use the adversarial examples to improve these systems and mitigate some negative impacts.
> >
> > In addition, the experimental settings have been widely adopted by previous methods like HLGA, TextHoaxer and LeapAttack.
> >
> > # Questions:
> > If I understand correctly, this work aims to employ the adversarial attack method to explore the system vulnerability and use the adversarial samples to further improve the system robustness. So my understanding is: The system developer should take the responsibility to examine/explore the system vulnerability and improve the robustness, rather than someone else out of the company that only has limited access to the model (only API access for example). Thus, why do you think the system developer, which is the core member of the model development, doesn't have access to the internal information (model parameters, gradients)?
> > In addition, I don't think following the unreasonable experimental settings in previous work is a strong argument.
> >
> >
> > # Author Response
> > (2) High-quality adversarial examples are essential
> >
> > In real world applications, low-quality adversarial examples can be easily detected by manual inspection or detection algorithms, which are meaningless to improve the system robustness. High-quality adversarial examples have higher semantic similarity and lower perturbation rate, which are easier to bypass manual inspection and detection algorithms, thus fulfilling the attack task. Furthermore, by utilizing high-quality adversarial examples to improve the original system, we can make the system more robust and safer.
> >
> > # Questions:
> > I'm very confused about the motivation of your attack method. Are you trying to show that the crafted adversarial samples can bypass the detection system to cause security issues, or you are trying to show that the constructed adversarial samples can be effective at improving the system robustness (by adversarial training maybe)?
> >
> > If you are the first case, please consider the case when attackers want to send some harmful content to the Twitter platform. They add many distracting and meaningless sentences to bypass the system. So do you think the Twitter company will spend a huge amount of money on doing human inspection on each message from every user to ensure the free of harmful meaning? Or you think there exist some concrete detection algorithms that can successfully detect such attack methods?
> >
> > If you are the second case, then you are trying to improve the system robustness. As I argue in the first question above, this responsibility should be taken by the system developers, which should have not only black-box access to the system. In this case, why should we use a black-box attack to craft adversarial samples?

---

> > > ### Comment · Reviewer_27hb · 2023-08-17
> > > **Philosophical Questions**
> > >
> > > As a respectful side note, these questions seem to touch on the philosophical aspects of the entire literature on black-box attacks, including both text and vision. While the exploration of the impracticability of (imperceptibility-oriented) black-box attacks is undoubtedly intriguing, it might warrant a separate paper as far as I can figure. Taking the vision black-box attack literature as an example, numerous studies have explored non-lp-norm or semantic imperceptibility, in the sense of "why adversarial examples should be lp-norm imperceptible." Yet, it is noteworthy that even the specific case of lp-norm bounded adversarial examples has not been solved. That being said, I am happy to follow up on the discussion, particularly in the text domain.

---

> > > > ### Comment · Reviewer_pVxP · 2023-08-17
> > > >
> > > > Hi Reviewer,
> > > >
> > > > Thanks for your note and explanations.
> > > >
> > > > I don't doubt the practicability of the black-box attacks. Black-box attacks are good simulation of real-world attack settings that we do not have gradient information about the target models. Actually, my point is, do we need a very complicated pipelien/algorithm to craft adversarial samples in this case? Do real-world attacks adopt such algorithm? If not, the proposed method is not a good simulation and doesn't have practical significance. I find the following papers that are relevant to our discussion, in both the NLP [2] and vision [1] domains. The basic point is: The real-world adversarial attacks don't need complicated pipeline to craft adversarial samples that are so-called high-quality and imperceptible, and the algorithm proposed in this work is query-inefficient compared to some heuristic attacks that real-world attackers are actually adopting, like the method proposed in [2].
> > > >
> > > > I list the practical issues as the weakness of this paper. But essentially, my major concern is about the research significance of this work, which is a fundamental limitation. Please correct me if I have some misunderstandings. From the practical side, as stated in my response to the authors, I don't find direct practical applications of the proposed algorithm (Please remind me if I miss something). From a theoretical point of view, the proposed algorithm also doesn't offer new insights.
> > > >
> > > >
> > > >
> > > >
> > > >
> > > >
> > > > [1] Stealthy Porn: Understanding Real-World Adversarial Images for Illicit Online Promotion. Kan Yuan, Di Tang, Xiaojing Liao, Xiaofeng Wang, Xuan Feng, Yi Chen, Menghan Sun, Haoran Lu, Kehuan Zhang et al
> > > >
> > > > [2] Why Should Adversarial Perturbations be Imperceptible? Rethink the Research Paradigm in Adversarial NLP. Yangyi Chen, Hongcheng Gao, Ganqu Cui, Fanchao Qi, Longtao Huang, Zhiyuan Liu, Maosong Sun

---

> > > > > ### Author Response · Authors · 2023-08-17
> > > > >
> > > > > **Dear Reviewer,**
> > > > >
> > > > > Thanks very much again for your effort. We would like to join in this discussion.
> > > > >
> > > > > Actually, we have carefully read the paper “Why Should Adversarial Perturbations be Imperceptible? Rethink the Research Paradigm in Adversarial NLP. EMNLP 2022.” when you point that in the original review.
> > > > >
> > > > > In Abstract of the above paper, it states that "In this paper, we rethink the research paradigm of textual adversarial samples in security scenarios". And in the section 3.1 (page 4) of the above paper, it states that "For example, when adversarial samples are adopted to augment existing datasets for adversarial training, we may aim for high-quality samples. Thus, the minor perturbations restriction is important." And in page 5 of the above paper, it clearly states that "Note that we don’t convey the meaning that the quality of adversarial samples is not important."
> > > > >
> > > > > Overall, that paper focuses in the security, but it seems not applicable in data augmentation, evaluation and so on. However, our work aims to generate high-quality adversarial samples which can be used to improve the robustness of the model.
> > > > >
> > > > > Here we list some potential application scenarios of our method.
> > > > >
> > > > > (1) Generating robust training data: Adversarial text examples can be used to generate robust training data. By introducing adversarial perturbations to clean text data, models can learn to be more robust and generalize better to real-world scenarios.
> > > > >
> > > > > (2) Natural language understanding: Adversarial text examples can help improve the natural language understanding capabilities of models. By crafting inputs that require nuanced comprehension or reasoning, models can be enhanced to better grasp complex language constructs.
> > > > >
> > > > > (3) Model robustness evaluation: Adversarial text examples are used to test the robustness of NLP models. By generating inputs that are intentionally designed to confuse or mislead the model, researchers can identify weaknesses in natural language processing algorithms.
> > > > >
> > > > > (4) Defense mechanism development: Adversarial text examples are used to develop and evaluate defense mechanisms for NLP models. These mechanisms aim to make models more resilient to adversarial attacks, ensuring their reliability in real-world applications.
> > > > >
> > > > > In addition, we would like to clarify our contributions. We have proposed a novel textual adversarial attack approach with a new substituting original words back procedure and a new optimization strategy, which performs much better than other strong baselines on 8 public datasets and 2 real-world APIs.
> > > > >
> > > > > In the revision, we will cite and discuss the differences between the above paper and our work.
> > > > >
> > > > > Thank you once again for your valuable time and effort.
> > > > >
> > > > > Best regards,
> > > > >
> > > > > Authors

---

> > > > > > ### Comment · Reviewer_pVxP · 2023-08-17
> > > > > >
> > > > > > Thanks for your detailed responses, and now I clearly understand the motivation for this work. I have one remaining question.
> > > > > >
> > > > > > Regarding the application scenarios listed above (e.g., generate new training data, NLU, robustness evaluation), why do we need to use the black-box setting, instead of some white-box attacks that are easier to craft adversarial samples? As I state in my response above, the application scenarios you list typically fall into the responsibility of the system developers, who not only have API access to the model.

---

> > > > > > > ### Author Response · Authors · 2023-08-18
> > > > > > >
> > > > > > > **Dear Reviewer,**
> > > > > > >
> > > > > > > **Thanks very much for your kind reply.**
> > > > > > >
> > > > > > > Both black-box and white-box adversarial attacks are important for a comprehensive understanding of the model vulnerabilities. These two settings are not opposite, but can work side-by-side.
> > > > > > >
> > > > > > > White-box attacks can provide insights into the inner workings of a model and guide improvements in its robustness from the perspective of system developers. Black-box attacks can simulate real-world scenarios where an attacker has limited knowledge about the target model from the perspective of malicious attackers.
> > > > > > >
> > > > > > > Using black-box attacks in certain application scenarios, even when white-box attacks might be easier to craft, can provide a more realistic and comprehensive assessment of the robustness and effectiveness of a model.
> > > > > > >
> > > > > > > Here are a few detailed reasons why black-box attacks are useful.
> > > > > > >
> > > > > > > (1) Realistic threat modeling: In many real-world scenarios, attackers may not have full access to the inner workings of a model or its architecture. Using black-box attacks better simulates the conditions under which an actual attacker might operate, leading to more accurate evaluation of the model's security and robustness.
> > > > > > >
> > > > > > > (2) Generalization of attacks: White-box attacks often rely on specific knowledge about the model's architecture, parameters, and gradients, which might not be available for all models or might change over time. Black-box attacks focus on finding vulnerabilities that are generalized across different models, making them more relevant for broader security assessments.
> > > > > > >
> > > > > > > (3) Assessment of real-world impact: Black-box attacks can help identify vulnerabilities that can be exploited in real-world scenarios where the attacker doesn't have complete knowledge of the system. This type of assessment is crucial for understanding potential risks and the impact of an attack on the system's users.
> > > > > > >
> > > > > > > (4) Third-party evaluations: When evaluating a model's performance and robustness, it's important to consider scenarios where the evaluator doesn't have access to the model's internals. This is especially relevant in scenarios where third-party auditors or researchers are conducting assessments without privileged access.
> > > > > > >
> > > > > > > (5) Continuous improvement: The study of black-box attacks can reveal new attack strategies and vulnerabilities that might not be apparent from white-box attacks alone. This ongoing exploration contributes to the development of more resilient and secure machine learning models.
> > > > > > >
> > > > > > > While white-box attacks might be easier to craft and can provide valuable insights into specific vulnerabilities, they often do not represent the full spectrum of potential threats that a model might face in the wild. A comprehensive evaluation strategy typically involves a combination of both black-box and white-box attacks to provide a well-rounded understanding of the model's robustness.
> > > > > > >
> > > > > > > **Thank you once again for your valuable time and effort, and we really appreciate your consideration.**
> > > > > > >
> > > > > > > Best regards,
> > > > > > >
> > > > > > > Authors

---

> > > > > > > > ### Comment · Reviewer_pVxP · 2023-08-18
> > > > > > > >
> > > > > > > > Dear authors,
> > > > > > > >
> > > > > > > > Thanks for your detailed explanations. The arguments offered are still somewhat far-fetched and somewhat inconsistent. For example, you said your work aims to generate high-quality adversarial samples which can be used to improve the robustness of the model in the previous response. But now you said why you consider black-box attacks is for realistic threat modeling and assessment of real-world impact. Is not this a security goal? Your previous response is "Overall, that paper focuses in security, but it seems not applicable in data augmentation, evaluation and so on. However, our work aims to generate high-quality adversarial samples which can be used to improve the robustness of the model.", which seems that your work is not focusing on security.

---

> > > > > > > > > ### Author Response · Authors · 2023-08-18
> > > > > > > > >
> > > > > > > > > **Dear Reviewer,**
> > > > > > > > >
> > > > > > > > > Thanks very much for your kind reply. We would like to further clarify and address your concerns.
> > > > > > > > >
> > > > > > > > > Textual adversarial samples play important roles in different domains of NLP research, including security, evaluation, explainability, and data augmentation [1].
> > > > > > > > >
> > > > > > > > > The primary focus of our work is to generate high-quality adversarial samples to assess and improve the robustness of models. In our opinions, the overall idea includes two steps, which seems clear and reasonable.
> > > > > > > > >
> > > > > > > > > (1) Using black-box attacks better simulates the conditions under which an actual attacker might operate, leading to more accurate evaluation of the model's robustness.
> > > > > > > > >
> > > > > > > > > (2) Leveraging the high-quality adversarial samples generated by black-box attacks to improve the robustness of the model.
> > > > > > > > >
> > > > > > > > > In summary, our work aims to enhance model performance by creating high-quality adversarial samples examples, and it has overlaps with security in the first step.
> > > > > > > > >
> > > > > > > > > The paper [1] mainly focuses on how to fool the model (security), and its main goal is to reconsider the attackers’ goals and reformalize the task of textual adversarial attack in security scenarios. But their generated adversarial samples seem not suitable for other tasks like data augmentation.
> > > > > > > > >
> > > > > > > > > We would like to express our appreciation for the research angle in the paper [1], and will provide a detailed discussion about the differences between [1] and our work in the revised version.
> > > > > > > > >
> > > > > > > > > **Thank you for the valuable time and effort to assist us in enhancing the quality of our manuscript.**
> > > > > > > > >
> > > > > > > > > **We really appreciate your consideration!**
> > > > > > > > >
> > > > > > > > > [1] Why Should Adversarial Perturbations be Imperceptible? Rethink the Research Paradigm in Adversarial NLP. EMNLP 2022.
> > > > > > > > >
> > > > > > > > > Best regards,
> > > > > > > > >
> > > > > > > > > Authors

---

> > > > > > > > > > ### Comment · Reviewer_pVxP · 2023-08-18
> > > > > > > > > >
> > > > > > > > > > Thanks for the detailed explanation. Although I'm not fully convinced, the arguments do make some sense to a certain extent. Given that the experimental conducted in this paper is solid and the algorithm is sound, I raise my score to 6.
> > > > > > > > > >
> > > > > > > > > > For the revision, I would encourage the authors to discuss clearly the motivation of the proposed algorithm, which lies in the fundamental core. Please also consider discussing and citing this paper [1] in the revision.
> > > > > > > > > >
> > > > > > > > > >
> > > > > > > > > >
> > > > > > > > > > [1] Stealthy Porn: Understanding Real-World Adversarial Images for Illicit Online Promotion. Kan Yuan, Di Tang, Xiaojing Liao, Xiaofeng Wang, Xuan Feng, Yi Chen, Menghan Sun, Haoran Lu, Kehuan Zhang et al

---

> > > > > > > > > > > ### Author Response · Authors · 2023-08-18
> > > > > > > > > > >
> > > > > > > > > > > Dear Reviewer,
> > > > > > > > > > >
> > > > > > > > > > > Thank you once again for your valuable time and effort. We deeply value your insightful and constructive comments.
> > > > > > > > > > >
> > > > > > > > > > > We will follow your suggestions to discuss and cite your mentioned paper in the revised version.
> > > > > > > > > > >
> > > > > > > > > > > Best regards,
> > > > > > > > > > >
> > > > > > > > > > > Authors

---

> > > > ### Author Response · Authors · 2023-08-17
> > > >
> > > > **Thanks very much for your attention.**
> > > >
> > > > Your participation greatly contributes to the enhancement of our paper’s quality and the enrichment of our contributions.
> > > >
> > > > And we deeply value the insightful comments provided by each reviewer, which play a pivotal role in refining our work.

---

> > > ### Author Response · Authors · 2023-08-17
> > >
> > > **Thanks very much for your kind reply. We really happy to discuss with you.**
> > >
> > > **Q1. About experimental settings (black-box attacks).**
> > >
> > > In our opinions, studying both black-box and white-box adversarial attacks is important for a comprehensive understanding of the vulnerabilities and defenses in AI security. These two settings are not opposite, but can work side-by-side.
> > >
> > > White-box attacks can provide insights into the inner workings of a model and guide improvements in its robustness from the perspective of system developers.
> > >
> > > Black-box attacks can simulate real-world scenarios where an attacker has limited knowledge about the target model from the perspective of malicious attackers.
> > >
> > > More specifically, the following points are why studying black-box attacks is essential, even when white-box attacks are considered.
> > >
> > > (1) Real-world scenarios: In many practical situations, attackers may not have full access to the target model's architecture, parameters, or training data. Black-box attacks reflect these scenarios, making them more realistic and applicable to real-world security concerns.
> > >
> > > (2) Adversarial robustness evaluation: Black-box attacks help researchers and practitioners assess the robustness of their models against a wider range of potential attacks. This evaluation is crucial to ensure that models perform well and securely in the presence of unknown adversaries.
> > >
> > > (3) Model agnostic defenses: Studying black-box attacks encourages the development of defenses that are not reliant on specific model details. This is important because defenses that only work under the assumptions of white-box attacks might be inadequate when dealing with more complex, real-world threats.
> > >
> > > (4) Continuous improvement: The study of black-box attacks can reveal new attack strategies and vulnerabilities that might not be apparent from white-box attacks alone. This ongoing exploration contributes to the development of more resilient and secure machine learning models.
> > >
> > > In addition, a lot of black-box attack methods have been widely-explored in various domains, such as computer vision [1-4], natural language processing [5-7], and even in the physical world [8-10], which also reflects the reasonability of this research direction to some extent.
> > >
> > > We will follow your suggestions to add these motivation explanations in the final version.
> > >
> > >
> > > **References**
> > >
> > > [1] Decision-based Black-box Attack Against Vision Transformers via Patch-wise Adversarial Removal. NeurIPS 2022.
> > >
> > > [2] Adv-Attribute: Inconspicuous and Transferable Adversarial Attack on Face Recognition. NeurIPS 2022.
> > >
> > > [3] QEBA: Query-efficient boundary-based blackbox attack. CVPR 2020.
> > >
> > > [4] Hopskipjumpattack: A query-efficient decision-based attack. IEEE S&P 2020.
> > >
> > > [5] LeapAttack: Hard-Label Adversarial Attack on Text via Gradient-Based Optimization. KDD 2022.
> > >
> > > [6] TextHoaxer: Budgeted Hard-Label Adversarial Attacks on Text. AAAI 2022.
> > >
> > > [7] Generating Natural Language Attacks in a Hard Label Black Box Setting. AAAI 2021.
> > >
> > > [8] Simultaneously Optimizing Perturbations and Positions for Black-box Adversarial Patch Attacks.IEEE TPAMI 2022.
> > >
> > > [9] Adversarial Sticker: A Stealthy Attack Method in the Physical World. IEEE TPAMI 2022.
> > >
> > > [10] Improving Transferability of Adversarial Patches on Face Recognition with Generative Models. CVPR 2021.
> > >
> > > **Q2. About high-quality adversarial examples.**
> > >
> > > We would like to further clarify that generating high-quality adversarial examples is essential for several reasons.
> > >
> > > (1) Robustness testing: High-quality adversarial examples serve as effective tools to test the robustness of machine learning models. By crafting these examples, researchers and practitioners can identify vulnerabilities and weaknesses in models' predictions and behavior, helping to improve their overall performance and reliability.
> > >
> > > (2) Improving models: High-quality adversarial examples are used during adversarial training, where models are trained on both clean and adversarial examples. This helps models learn to be more robust and resilient to adversarial perturbations.
> > >
> > > (3) Advancing research: Crafting high-quality adversarial examples drives advancements in adversarial machine learning research. Researchers develop new techniques and algorithms to generate more challenging and realistic adversarial examples, pushing the boundaries of our understanding and defenses.
> > >
> > > In addition, we think that both black-box and white-box adversarial attacks are important for a comprehensive understanding of the vulnerabilities and defenses in AI security. The detailed reasons are explained in the first answer.
> > >
> > >
> > > **Thank you once again for your valuable time and effort.**

---

### Official Review · Reviewer_RbiV · 2023-07-04

**Soundness:** 3 good
**Presentation:** 2 fair
**Contribution:** 3 good
**Rating:** 5
**Confidence:** 3

**Summary:**

- The authors consider a problem of black-box hard-label adversarial attack, which adversarially perturbs the input text while having access only to the output label of the victim model.
- They propose HQA-Attack method which can successfully perturb the input text into the adversarial text with higher semantic similarity and lower perturbation rate compared to the baseline hard-label adversarial attack methods on text data under a tight query budget.
- The process can be summarized into three components:
1. Random initialization to find an adversarial example.
2. Sequentially substitutes original words back as many as possible, beginning from the adversarial example found in step 1.
3. Further optimization step to improve semantic similarity and perturbation rate.
- The experimental results show the HQA-Attack's superiority in hard-label adversarial attacks.

**Strengths:**

- The authors conduct experiment on various datasets and victim models containing commercial classification system such as google cloud and alibaba cloud.
- The second step of HQA-Attack, which greedily substitute original words back, is very simple and intuitive.


**Weaknesses:**

- The attack space is limited. HQA-Attack seems to construct a word substitution candidate set of a word without considering any knowledge from the sentence.
- The justification of the third step seems poor. I think more theoretical or empirical evidence that supports the necessity of the third step should be added.

**Questions:**

- Soft-label adversarial attack methods such as TextFooler, BERT Attack, and BAE considers the word substitution set which depends on the whole sentence. In my understanding, HQA-Attack can be extended to more flexible attack spaces equipped with sentence-dependent word substitution sets. Can you provide some experimental results on this setting?
- The third step of HQA-Attack is not intuitive to me. Can you provide the theoretical or empirical results that show the importance of the third step? For example, I want to see an ablation result with and without the third step.
- In BBA, they first find the adversarial example with Bayesian Optimization and conduct a "post-optimization" process which further optimizes near the adversarial example to reduce the perturbation size based on trained Gaussian Process (GP). While this method is solving the soft-label adversarial attack problem, the idea of using GP to consider previous evaluation history seems still valid in the hard-label attack setting. Can you provide a comparison of HQA-Attack with a simple GP-based hard-label attack?

[TextFooler] Is BERT Really Robust? Natural Language Attack on Text Classification and Entailment, Di Jin et al., AAAI 2020

[BERT Attack] BERT-ATTACK: Adversarial Attack against BERT using BERT, Linyang Li et al., EMNLP 2020

[BAE] BAE: BERT-based Adversarial Examples for Text Classification, Siddhant Garg et al., EMNLP 2020

[BBA] Query-Efficient and Scalable Black-Box Adversarial Attacks on Discrete Sequential Data via Bayesian Optimization, Deokjae Lee et al., ICML 2022




**Limitations:**

The authors provide some broader impacts and limitations in Appendix F.

---

> ### Author Rebuttal · Authors · 2023-08-09
>
> **For Reviewer RbiV.**
>
> Thanks very much for your valuable and positive comments.
>
> **1. About the experiments with the sentence-dependent word substitution set.**
>
> (1) For fair comparison, we follow the previous works HLGA, TextHoaxer and LeapAttack to generate 50 synonyms for each word by using the counter-fitting word vector.
>
> (2) We have followed your comments to conduct experiments with the sentence-dependent word substitution set. Specifically, we modify our method with BERT-based synonyms to attack the WordCNN model. The results are as follows.
>
> | Dataset | Sim(%) | Pert(%) |
> | :-----: | :----: | :-----: |
> |   MR    |  73.4  | 11.856  |
> |   AG    |  81.3  | 10.798  |
> |  Yahoo  |  82.6  |  6.071  |
> |  Yelp   |  87.9  |  6.249  |
> |  IMDB   |  92.2  |  3.740  |
>
> From the above results and Table 2 of the original paper, it can be seen that BERT-based synonyms are competitive with counter-fitting word vector based synonyms. We will further attempt to consider these two types of synonyms simultaneously to improve the method in the future.
>
> **2. Ablation study of the third step.**
>
> In the original paper, we have shown the ablation study of the third step in Table 6. Here we list the results (Sim(%)/Pert(%)) as follows.
>
> | Dataset | w/o Optimizing | HQA-Attack  |
> | ------- | -------------- | ----------- |
> | MR      | 74.2/11.673    | **75.1/11.224** |
> | AG      | 79.3/11.305    | **82.7/8.831**  |
> | Yahoo   | 79.7/7.445     | **82.4/6.132**  |
> | Yelp    | 85.2/7.883     | **87.8/6.312**  |
> | IMDB    | 92.2/3.923     | **93.1/3.285**  |
>
> From the results, it can be seen that the step of optimizing the adversarial example is necessary for HQA-Attack.
>
> **3. About the comparison of HQA-Attack with a simple GP-based hard-label attack.**
>
> We have modified the BBA method to construct a simple GP-based hard-label attack model by replacing the prediction score with the hard label (0 or 1). We take the WordCNN as the victim model, and the results are as follows.
>
> | Dataset | Sim(%) | Pert(%) |
> | :-----: | :----: | :-----: |
> |   MR    |  63.6  | 14.946  |
> |   AG    |  72.3  | 14.573  |
> |  Yahoo  |  71.6  |  8.931  |
> |  Yelp   |  78.5  |  9.913  |
> |  IMDB   |  89.6  |  4.421  |
>
> From the above results and Table 2 of the original paper, it can be seen that HQA-Attack still performs better than the simple GP-based hard-label attack model. The possible reason is that the acquisition function used in GP heavily relies on the concrete prediction score of the victim model, but in the hard-label setting only predicted labels are accessible.

---

> > ### Comment · Reviewer_RbiV · 2023-08-17
> >
> > My intention in the third question was to compare only the "shrinking method" part of the HQA attack and BBA.
> >
> > Your response to the other questions was very helpful.
> >
> > I decide to maintain my rating on this paper.
> >
> > Please consider appending these results to the revised version.
> >
> > Thank you for your response!

---

> > > ### Author Response · Authors · 2023-08-18
> > >
> > > Dear Reviewer,
> > >
> > > Thanks very much for your insightful and valuable comments.
> > >
> > > We will attempt to explore the "shrinking method" part of the HQA attack and BBA. All new results will be added in the revised version.
> > >
> > > Best regards,
> > >
> > > Authors

---

### Official Review · Reviewer_27hb · 2023-07-05

**Soundness:** 3 good
**Presentation:** 3 good
**Contribution:** 3 good
**Rating:** 6
**Confidence:** 4

**Summary:**

This paper proposes a new hard-label black-box attack on textual data that aims to improve the performance on semantic similarity and perturbation rate. The main novel techniques are word back-substitution and a new strategy for optimizing the adversarial example. Evaluations on 5 classification tasks, 3 inference tasks, and 2 online APIs show that the proposed attack obtains better quality than 3 previous attacks. Evaluations also include the attack's convergence, the performance on robust models, and several ablation studies.

**Strengths:**

### Originality

* **Novel attack.** The attack techniques are overall novel to my knowledge. The proposed strategy to optimize quality is new.

### Quality

* **Solid methodology.** The overall attack design is solid, and the new optimizing strategy is well-developed.
* **Comprehensive evaluations.** The experiments are comprehensive. I appreciate the several tasks and the inclusion of robust models and ablation studies.

### Clarity

* **Good presentation.** The paper is generally well-written and the methodology is largely easy to follow with some focus.

### Significance

* **Improved performance regarding quality.** It is good to see that the proposed optimization strategy can improve the adversarial example's quality in all settings. The convergence behavior is also better than previous attacks.


**Weaknesses:**

### Originality

**Q1: Unclear novelty in the word back-substitution component.**

I was a bit confused by the contribution in Section 4.2. From L149-153, it seems that previous attacks only compute the similarity gain once and hence determining the replacement order, so the initially computed order is inaccurate after the adversarial example has applied any replacement. However, isn't this the same case for Equation (3) and Algorithm 1, where the order is computed once (Algorithm 1, L2-7), and then the attack applies replacements one by one (Algorithm 1, L10-17)? Basically, it is unclear how the proposed method is different from previous attacks, and how it would be able to solve this problem.

If there is indeed a difference, what if previous attacks do recompute this order at each iteration? Would they have better performance, or is there an inherent challenge in their methodology? From the current presentation I could not see any challenge, as this computation is generally offline and does not need model queries. In other words, in this part of the question, it is hard to evaluate the novelty between "naively recomputing the order" and "recomputing the order with the proposed strategy."

### Quality

**Q2: The qualitative results are somewhat insufficient.**

While the quantitative results in Section 5 are good and promising, the current qualitative results are a bit insufficient for demonstrating the improved quality. Currently, only three sentences are provided in the supplementary, and it would be hard to tell if they have covered the general sense of improved quality. Since this paper's main claim is the improved quality of adversarial examples, it might be important to have more examples of the adversarial examples and, more importantly, the comparisons between different attacks. It is not intuitive to tell the difference between 5-10% of similarity or perturbation rate improvement. For example, would there be an observable quality improvement if the semantic similarity was improved by 10%? Note that I have no concerns with the chosen metrics, but the paper can be strengthened if more qualitative results are provided.

**Q3: The choice of $k$.**

At L259, the number of synonyms sampled for direction estimation is set to $k=5$. I am curious if this value would be too small, given that there might be tens of synonyms. I also noticed that the results are constant regarding $k\in\{5, 10, 15\}$ so $k=5$ seemed to be sufficient. Is there any explanation for why this is sufficient? In this case, it might be informative to add results for $k\in[1, 4]$ where the number of samples is insufficient.

### Clarity

**Q4: Confusion in determining the optimizing order.**

It is not very straightforward why Section 4.3.1 determines that the most similar word pair get optimized first. In Equation (6), the word $w_i$ gets sampled with a higher probability if $d_i$ is small, meaning the cos similarity between (the embeddings of) $w_i$ and $w^\prime_i$ is high. Can you clarify why it starts from similar words (rather than non-similar words), and replace such similar words with synonyms that again aim to improve semantic similarity?

**Q5: Confusion in the final updating step.**

In L224-225, why the replacement word is chosen from $S(w_i)$, yet the similarity was compared pertaining $\bar{w}_i$?

**Minor Points:**
*  Missing specification of the synonyms set. It is unclear how the synonyms were generated and how large it is (relevant to Q3).
* The methodology description in Section 4.3.2 is a bit convolved.


### Significance

**Q7: Used the same random initialization as previous attacks.**

While the improvement in quality is promising, and this paper legitimately focuses on improving the quality, the task's performance is not reduced more than other attacks because they share the same initialization (source of the adversary). I am not penalizing for this point, but this is one aspect that limits the significance of this paper (compared with attacks that might also improve the attack success rate).

**Questions:**

I am willing to raise my score if Q1-Q2 (major) and Q3-Q5 (intermediate) are sufficiently clarified.

---

> ### Author Rebuttal · Authors · 2023-08-09
>
> **For Reviewer 27hb.**
>
> Thanks very much for your valuable and positive comments.
>
> **1. About the word back-substitution component.**
>
> (1) For the word back-substitution component, HQA-Attack is different from previous methods. Specifically, as shown in Alg. 1 (Appendix A), after replacing one original word successfully (Line 12-Line 13), our method can make the $flag$ = False (Line 14) and break out of the current loop (Line 15), where the current loop is from Line 10 to Line 17. Furthermore, as the $flag$ = False, our method will continue the outer loop and recompute the order, where the outer loop is from Line 1 to Line 21. By replacing the best original word in each iteration, HQA-Attack can make the intermediate generated adversarial example have the highest semantic similarity with the original example.
>
> (2) We have conducted experiments to verify the effectiveness of our proposed word back-substitution strategy. Specifically, we use our proposed word back-substitution strategy to replace the original word back-substitution strategy in LeapAttack to attack the WordCNN model. (Original) means the original method, and (Improved) means the method with our proposed word back-substitution strategy. The results (Sim(%)/Pert(%)) are shown as follows.
>
>
> | Dataset |  LeapAttack (Original) | LeapAttack (Improved) |
> | -------- | -------------------- | -------------------- |
> | MR       | 63.2/14.016          | **65.3/13.809**          |
> | AG       | 72.0/12.827          | **74.8/12.013**          |
> | Yahoo    | 74.3/7.842           | **74.6/7.623**           |
> | Yelp     | 80.1/8.816           | **81.5/8.120**           |
> | IMDB     | 89.7/3.886           | **90.8/3.796**           |
>
> The results show that our proposed word back-substitution strategy can improve the semantic similarity and reduce the perturbation rate successfully. We will add these results in the final version.
>
> **2. About the qualitative results.**
>
> We have conducted the human evaluation experiments on the Bert model using HLGA, TextHoaxer, LeapAttack and HQA-Attack for the MR and IMDB datasets. Specifically, for each dataset, we first randomly select 50 original samples, and use each adversarial attack method to generate the corresponding 50 adversarial examples respectively. Then we ask 10 volunteers to annotate the class labels for these samples, and calculate the average classification accuracy (Acc) for each method. Intuitively, if the accuracy is higher, it means that the quality of the generated adversarial examples is better. The detailed Acc(%) results are as follows.
>
> | Dataset | HLGA | TextHoaxer | LeapAttack | HQA-Attack |
> | ------- | ---- | ---------- | ---------- | ---------- |
> | MR      | 82.6 | 84.8       | 84.0       | **87.4**   |
> | IMDB    | 84.4 | 85.4       | 85.0       | **88.6**   |
>
> The results show that the adversarial examples generated by HQA-Attack are more likely to be classified correctly, which further verifies the superiority of HQA-Attack in preserving the semantic information. We will follow your comments to add these results in the final version.
>
> **3. About the choice of $k$.**
>
> We further conduct experiments for $k \in [1,4]$ when attacking WordLSTM, the results (Sim(%)/Pert(%)) are shown in the following table.
>
> | $k$\Dataset | MR          | AG          | Yahoo      | Yelp       | IMDB       |
> | ---------- | ----------- | ----------- | ---------- | ---------- | ---------- |
> | $k=1$        | 73.6/11.551 | 74.9/11.257 | 72.7/7.231 | 86.4/6.317 | 90.4/3.601 |
> | $k=2$        | 73.7/11.504 | 75.0/11.286 | 73.1/7.050 | 86.5/6.314 | 90.7/3.491 |
> | $k=3$        | 73.9/11.635 | 75.2/11.491 | 73.4/6.901 | 86.8/6.279 | 90.9/3.421 |
> | $k=4$        | 74.0/11.499 | 75.4/11.364 | 73.8/6.764 | 87.1/6.161 | 91.2/3.311 |
>
> Combining with the results $k =5, 10, 15$ in Table 2 of the supplementary material, we can find that $k=5$ is sufficient.
>
> **4. About the confusion in the optimizing order.**
>
> In the procedure of substituting the original words back, if the changed word has a high semantic similarity with the original word but is not replaced by the original word, it indicates that this word position is important for the adversarial example. This circumstance can be summarized as if the semantic similarity between the changed word and the original word is higher, the word position is more important. Based on the observation, we use the strategy in Section 4.3.1 to determine the optimizing order.
>
> **5. About confusion in the final updating step.**
>
> (1) For fair comparison, we exactly follow the previous works HLGA, TextHoaxer and LeapAttack to choose the replacement word from $S(w_i)$.
>
> (2) To guarantee the semantic difference between the replacement word and the original word is always small, we choose the replacement word from $S(w_i)$.
>
> **6. About the minor points.**
>
> (1) For fair comparison, we follow the previous works HLGA, TextHoaxer and LeapAttack to generate 50 synonyms for each word by using the counter-fitting word vector. We will add these explanations in the final version.
>
> (2) We will polish the methodology description in Section 4.3.2 in the final version.
>
> **7. About the initialization.**
>
> (1) In the hard-label scenario, attackers can only access the model output labels. Therefore, in most existing hard-label settings, no matter images or texts, random initialization is the most commonly used method to obtain the initial adversarial example.
>
> (2) The initialization strategy is really a good research direction, one possible idea is to calculate the importance score of each word for the final prediction or evaluate the impact of the replacement of each word on the final prediction. We will further explore this direction in the future.

---

> > ### Comment · Reviewer_27hb · 2023-08-12
> >
> > Thanks for the response, my questions are clarified and I have raised my score to 6 accordingly.
> >
> > Below is a minor follow-up.
> >
> > **Q2: Qualitative results.**
> >
> > My initial thought was asking for more text examples like Table 3 in the appendix, but the added human evaluation is also very interesting. Given this result, it is suggested to highlight the definition of "high-quality" more (right now it is hidden in L75-77). For example, compared with "higher semantic similarity," "higher human imperceptibility" (or a similar notion) sounds more intuitive (to me), in the sense that it becomes harder for a human to classify adversarial examples differently from clean samples. To demonstrate this better, the added results should include a baseline of clean samples (also act as the optimal-quality attack).

---

> > > ### Author Response · Authors · 2023-08-13
> > >
> > > Thanks very much for your kind reply.
> > >
> > > **About qualitative results.**
> > >
> > > (1) We will add more text examples like Table 3 in the appendix.
> > >
> > > (2) We have conducted the experiments about the baseline of clean samples, and the corresponding accuracy scores on the MR and IMDB datasets are 94.2% and 93.6% respectively. We will add these results in the final version.

---

### Official Review · Reviewer_gEht · 2023-07-07

**Soundness:** 3 good
**Presentation:** 3 good
**Contribution:** 2 fair
**Rating:** 5
**Confidence:** 4

**Summary:**

This paper proposes a novel and effective methods for black-box textual adversarial attacks. The proposed framework focuses on maximizing the similarity of the adversarial texts and the original texts through shrinking the perturbation. In the experiments, the authors prove that the proposed methods can significantly improve the quality of generated adversarial examples according to the metrics of text similarity and perturbation rate. The authors also conduct comprehensive ablation studies to show the effectiveness of different module in the proposed framework.

**Strengths:**

1. This paper accurately points out the limitations of the previous work on textual adversarial attacking, and it proposes a simple and effective methods to improve the quality of generated adversarial examples.
2. The experiments are exhaustive and convincing. The selection of evaluation benchmarks is comprehensive, and it also evaluate on real-world APIs, which further illustrate the practicability of the proposed methods. According to the results, the proposed methods can significantly improve the quality of adversarial examples. In addition, the ablation study also interprets the effectiveness of different module of the attacking framework.

**Weaknesses:**

1. It lacks human evaluation to evaluate the quality of the generated adversarial examples. The human evaluation is claimed critical in previous textual adversarial works, since some slight perturbations that changes the semantic of the original texts may not be reflected through automatic evaluation such as text similarity and perturbation rate. (The authors have added the human evaluation experiments.)
2. It would be helpful for us to better understand the mechanism of adversarial attacks if the authors can add some analysis of why the adversarial examples can successfully full the models / what triggers the wrong predction / whether different victim models make different mistakes or same mistakes.

**Questions:**

1. The authors should add human evaluation experiments about the quality of generated adversarial examples.


**Limitations:**

The authors should address the limitation of the proposed framework in the conclusion section.

---

> ### Author Rebuttal · Authors · 2023-08-09
>
> **For Reviewer gEht.**
>
> Thanks very much for your insightful and positive comments.
>
> **1. Add human evaluation experiments.**
>
> We have conducted the human evaluation experiments on the Bert model using HLGA, TextHoaxer, LeapAttack and HQA-Attack for the MR and IMDB datasets. Specifically, for each dataset, we first randomly select 50 original samples, and use each adversarial attack method to generate the corresponding 50 adversarial examples respectively. Then we ask 10 volunteers to annotate the class labels for these samples, and calculate the average classification accuracy (Acc) for each method. Intuitively, if the accuracy is higher, it means that the quality of the generated adversarial examples is better. The detailed Acc(%) results are as follows.
>
> | Dataset | HLGA | TextHoaxer | LeapAttack | HQA-Attack |
> | ------- | ---- | ---------- | ---------- | ---------- |
> | MR      | 82.6 | 84.8       | 84.0       | **87.4**   |
> | IMDB    | 84.4 | 85.4       | 85.0       | **88.6**   |
>
> The results show that the adversarial examples generated by HQA-Attack are more likely to be classified correctly, which further verifies the superiority of HQA-Attack in preserving the semantic information. We will follow your comments to add these results in the final version.
>
> **2. It would be helpful for us to better understand the mechanism of adversarial attacks if the authors can add some analysis of why the adversarial examples can successfully fool the models/what triggers the wrong prediction/whether different victim models make different mistakes or same mistakes.**
>
> (1) We would like to attempt to analyze our method from the perspective of decision boundary. Specifically, HQA-Attack consists of three steps. First, HQA-Attack generates an adversarial example by initialization, which means that the adversarial example is outside the decision boundary associated with the original true label. Second, HQA-Attack deals with the adversarial example by substituting original words back, which means that the adversarial example is getting closer to the decision boundary, thus improving the semantic similarity and reducing the perturbation rate. Third, HQA-Attack further optimizes the adversarial example along the direction that can increase the semantic similarity, which means the adversarial example is further approaching the decision boundary. The first step determines whether the adversarial example can fool the model and trigger the wrong prediction. The last two steps determine the quality of the adversarial example.
>
> (2) In general, different victim models have different decision boundaries, but they share the same label space. Therefore, one adversarial example can cause that different victim models make different mistakes or same mistakes. We list two adversarial examples generated by HQA-Attack on the AG dataset in the following table, and the substitute words are indicated by parentheses.
>
> | Adversarial Example                                          | True Label | BERT Prediction    | WordCNN Prediction  | WordLSTM Prediction |
> | ------------------------------------------------------------ | ---------- | -------- | -------- | -------- |
> | Witnesses to confront cali cartel ~~kingpin~~ (**punisher**) thirteen years into their probe , u.s. investigators have assembled a team of smugglers , accountants and associates to testify against colombian cartel ~~kingpin~~ (**studs**) gilberto rodriguez orejuela. | World      | Sports   | World    | Business |
> | Boys ' cured ' with gene therapy gene therapy can cure ~~children~~ (**enfants**) born with a condition that knocks out their natural defences against infection , mounting evidence shows. | World      | Sci/Tech | Sci/Tech | Sci/Tech |
>
> According to the results, we can get that the first example shows that different victim models can make different mistakes, and the second example shows that different victim models can make the same mistake.
>
> (3) We will follow your comments to add these explanations in the final version.

---

> > ### Comment · Area_Chair_sGUZ · 2023-08-19
> > **Thanks for your detailed response.**
> >
> > Dear authors,
> >
> > Thanks for your detailed response. I think your response has addressed some of the reviewer's concerns.
> >
> > Best,
> >
> > AC

---

> > > ### Author Response · Authors · 2023-08-19
> > >
> > > Dear AC,
> > >
> > > We would like to thank you and each reviewer for devoting the valuable time and effort to assist us in enhancing the quality of our manuscript.
> > >
> > > Best regards,
> > >
> > > Authors

---

### Author Rebuttal · Authors · 2023-08-09

**For everyone.**

We would like to thank everyone for devoting the valuable time and effort to assist us in enhancing the quality of our manuscript.

We have provided the detailed responses for each review, and we really appreciate your consideration.

---

> ### Comment · Area_Chair_sGUZ · 2023-08-13
> **Thanks for your insightful reviews**
>
> Dear reviewers,
>
> Thanks for your insightful reviews. It appears that we have received a little bit of divergent reviews. While most reviewers are positive toward the paper, some reviewers have raised questions regarding the contributions and experiments. Now that the authors have provided the detailed responses for each review, we kindly request those of you who haven't read the response to take a moment to review these materials and determine if they effectively address the concerns you raised.
>
> Best,
>
> AC

---

### Decision · Program_Chairs · 2023-09-21

**Decision:**

Accept (poster)

**Comment:**

The paper proposes a new hard-label black-box attack on textual data that aims to improve the performance in semantic similarity and perturbation rate. Experiments on various datasets and victim models verify that the proposed methods can significantly enhance the quality of generated adversarial examples, according to the metrics of text similarity and perturbation rate.

Initially, we received slightly divergent reviews. While the majority of reviewers responded positively to the paper, some raised questions regarding its contributions and experiments. Subsequently, the authors' responses successfully addressed specific aspects of the reviewers' concerns, prompting several of them to raise their scores above the threshold. Consequently, I would like to suggest acceptance.